



# "Norkyst" version 3: the coastal ocean forecasting system for Norway

Kai Håkon Christensen[1,3], Jon Albretsen[2], Lars Asplin[2], Håvard Guldbrandsen Frøysa[2], Yvonne Gusdal[1], Silje Christine Iversen[1], Mari Fjalstad Jensen[2], Ingrid Askeland Johnsen[2], Nils Melsom Kristensen[1], Pål Næverlid Sævik[2], Anne Dagrun Sandvik[2], Magne Simonsen[1], Jofrid Skarðhamar[2], Ann Kristin Sperrevik[1], and Marta Trodahl[1,†]

[1]Norwegian Meteorological Institute, Oslo, Norway
[2]Institute of Marine Research, Bergen, Norway
[3]University of Oslo, Oslo, Norway
[†]Now at Equinor, Stavanger, Norway

**Correspondence:** Kai H. Christensen (kaihc@met.no)

**Abstract.** We describe the operational forecasting system "Norkyst", now in version 3, which is used for predicting the ocean circulation along the coast of mainland Norway and in the fjords. The forecasting system is based on the Regional Ocean Modeling System (ROMS), and has sub-kilometric horisontal resolution to resolve mesoscale features. Here we describe the basic configuration and report verification statistics of unconstrained model runs. The main features of the circulation and hydrography, including seasonal variations, are well represented in Norkyst v.3, making the forecast system suitable for its

intended use as an open service for users in public or private sectors such as aquaculture, fishery, shipping, research, consulting, environmental management, and others who needs detailed predictions of the physical state of the Norwegian coastal ocean.

## 1   Introduction

Mainland Norway has a very long coastline which stretches from the Skagerrak in the south to the Barents Sea in the north

(Fig. 1). As the name implies, the Norwegian coast is a gateway to the Arctic. The main ocean currents in this region are (i) the Norwegian Atlantic Current (NwAC), an extension of the North Atlantic Current that carries warm, saline water to the Arctic Ocean, and (ii) the Norwegian Coastal Current (NCC), originating in the Skagerrak from mixed waters of the North Atlantic, North Sea, Baltic Sea, and river run-off, flowing along Norway's coast into the Barents Sea (e.g., Sætre, 2007). The coastline itself is rather complex, with many long and deep fjords and more than 200.000 large and small islands. The coastal waters are

often strongly stratified and the shelf circulation is dominated by energetic eddies.

There are large variations in the prevailing weather conditions along the coast. The coastal climate in the south-eastern parts of Norway is temperate while it is boreal in the north. Seasonal cycles in radiative forcing also have large amplitudes, with about one third of Norway being north of the polar circle. The day length variations cause the pronounced spring blooms that inspired Sverdrup (1953) to develop his "critical depth theory", continuing a long tradition of scientific investigations, starting

with Helland-Hansen and Nansen (1909), linking the ocean physics to its biology to better understand the behaviour of the large





fish stocks that have always been so important to the Norwegian economy. Today, the fisheries and aquaculture are important sectors, as is the offshore energy sector, which is still dominated by oil and gas production.

Robust and accurate forecasts are needed for many purposes such as decision support (search-and-rescue, ship drift, oil spill mitigation, etc.), operations planning and execution, and for estimating oceanic transports of plankton or pollutants. The complex topography means that we need high spatial resolution in our coastal ocean forecasting system to properly resolve the dynamics. The Norwegian national operational forecasting system "Norkyst" was established in 2011 to meet these needs (Albretsen et al., 2011), put into operations in 2012, and saw a major upgrade in 2019. The forecasting system is based on the Regional Ocean Modeling System (ROMS, Shchepetkin and McWilliams, 2005; Haidvogel et al., 2008).

The present configuration of Norkyst is version 3. The main differences compared with earlier versions are (i) extended domain, (ii) updated bathymetry and landmask, (iii) improved numerics, (iv) increased number of vertical layers, (v) improved atmospheric forcing, and (vi) improved river discharge forcing. Norkyst is the national complement to the Copernicus Marine Service, providing dynamically downscaled information as it is nested into regional modeling systems from the Arctic and Baltic forecasting centres.

This paper describes the basic "Norkyst" configuration of ROMS, and we also present verification statistics from a long hindcast, demonstrating the ability of the unconstrained forecast system to represent the dynamics of the Norwegian coastal ocean. Norkyst data are currently offered as (i) five day forecasts and (ii) as a continuous hindcast archive going back to 2012, and which is being updated quarterly and is currently covering up to and including December 2024. The outline of the paper is as follows: In Sec. 2 we describe the model domain and our specific configuration of ROMS; in Sec. 3 we describe the external forcing (riverine forcing, atmospheric forcing, lateral boundary conditions); in Sec. 4 we present verification scores from a free model run covering a period of over ten years; in Sec. 5 we describe the operational implementation and forecast data dissemination; and, finally, Sec. 6 contains an outline of ongoing developments and a few concluding remarks.

## 2 Dynamical core

Norkyst is based on the Rutgers version of ROMS (https://github.com/myroms), with a model domain on a curvilinear rotated polar stereographic grid, using stretched vertical coordinates. The physical core is based on the hydrostatic approximation. There are four control variables primarily associated with the slow baroclinic mode: the depth dependent horisontal velocities $(u, v)$ in addition to salinity, $S$, and potential temperature, $T$. Three variables are primarily associated with the fast barotropic mode: the depth-averaged horisontal velocities, $(\bar{u}, \bar{v})$ and the surface coordinate, $\zeta$. The fast and slow modes are coupled through a split-explicit time stepping scheme (Shchepetkin and McWilliams, 2005). Although some Norwegian fjords can be partially or entirely covered in sea ice during winter (O'Sadnick et al., 2020), we have not activated any inbuilt sea ice model components or use any sea ice model coupling in Norkyst. Instead we have activated a simple parameterization available in ROMS that limits the cooling of the surface layer to avoid having temperatures below the freezing point. A dedicated ocean-sea ice forecasting system for the Barents Sea and the areas around Svalbard, which also covers the northernmost part of mainland Norway, is based on a coupling between ROMS and the sea ice model CICE (Röhrs et al., 2023).



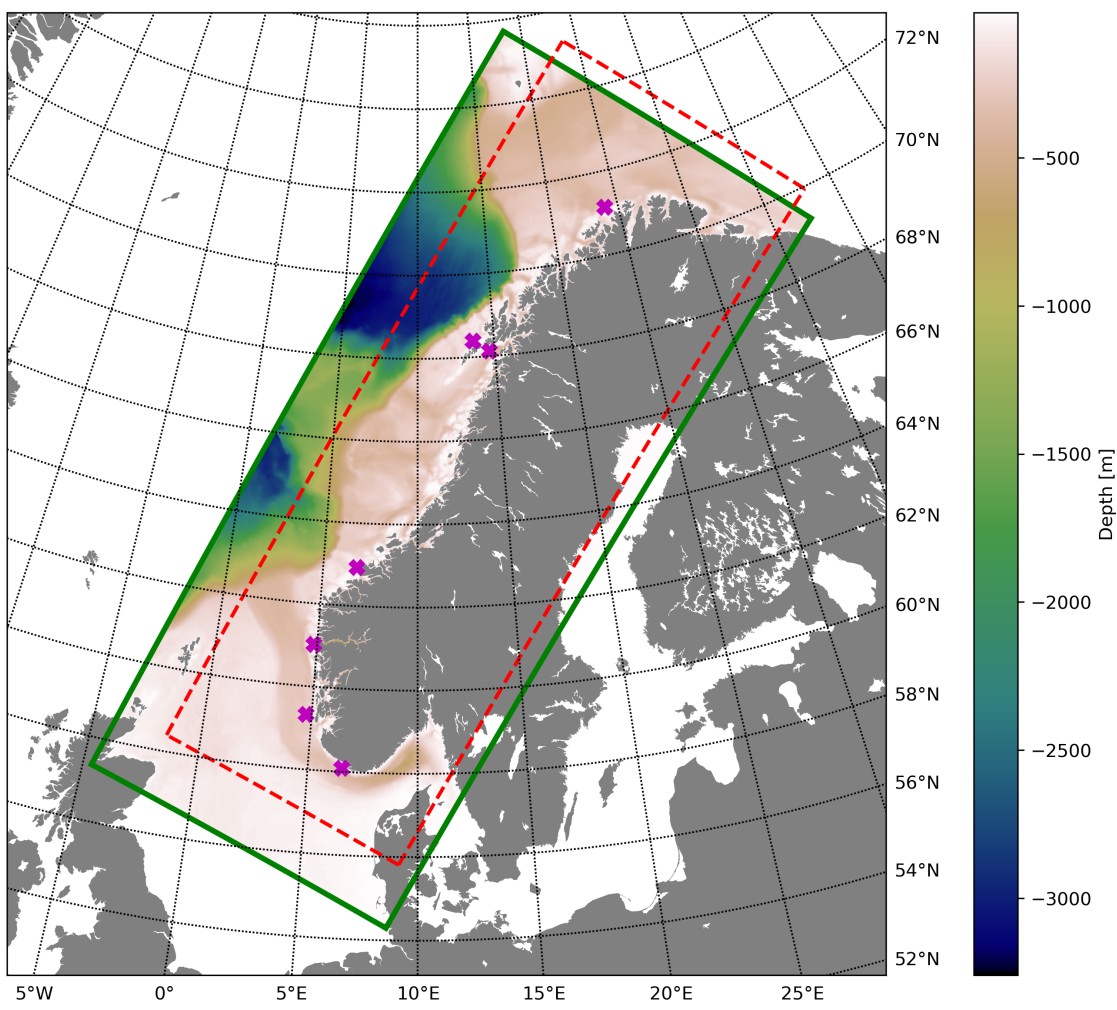

**Figure 1.** The Norkyst v.3 model domain and bathymetry (green solid line). The outline of the previous version of Norkyst is also depicted here (red dashed line). The v.3 model domain stretches from the North Sea in the south to the Barents Sea in the north. A pronounced shelf break separates the Norwegian coastal ocean from the Norwegian Sea in the west. The slope is in places very steep, especially outside the Lofoten archipelago and northwards between the 68th and 72nd parallels. The magenta crosses mark the positions of the coastal stations where long term hydrographical measurements are collected (see Ch. 4).





For this new version, we have made extensive tests of bathymetry smoothing methods, model minimum depth values, advection schemes, turbulence parameters, and so on, to the best of our ability choosing settings and values that are consistent with the dynamics while still producing a stable and robust modeling system without significant biases. A detailed description of all these tests are outside the scope of this paper, but interested readers are encouraged to contact the authors directly if they wish for more in-depth information on the choice of specific parameters. The final configuration was chosen based on overall verification statistics of currents and hydrography from unconstrained model simulations (see Ch. 4).

## 2.1 Model grid

The horisontal resolution of Norkyst is approximately 800 m. ROMS uses C grid staggering, and the domain size is 2747 x 1148 horisontal grid points. The maximum model depth is 3257 m. The bathymetric data are based on a combination of EMODnet data and high resolution data from the Norwegian Mapping Authority. For numerical stability, the depth matrix is smoothed using a Laplacian filter, and a minimum depth of 10 m. Alternative approaches, such as volume-conserving smoothing methods were also tested, but did not yield satisfactory results. These methods made it more difficult to control pressure gradient errors. Norkyst has 40 vertical layers. We prioritise high resolution in the uppermost layers due to the importance of near-surface flows for operational services, which comes at the expense of somewhat reduced ability to resolve the dynamics at intermediate depths and close to the sea floor. Typical vertical resolution close to the surface varies from 0.1 to 1.0 m where the depth is 10 or 1000 m respectively, while the thickness of the bottom layer ranges from 0.4 m at 10 m depth to 23.3 m at 1000 m depth. About 42% of the grid is land. In places where the resolution is inadequate to properly describe the coastline, we have either opened up or closed sounds and passages, aiming for as realistic overall circulation patterns as possible.

## 2.2 Time stepping and advection schemes

In Norkyst, we use a baroclinic time step of 40 s and a barotropic time step of 2 s. The main challenge with regards to stability is not associated with horisontal resolution, but with occasional large vertical velocities in regions of strong convergence (e.g. at the Kattegat-Skagerrak front), which in turn leads to violations of the CFL criterion in the vertical tracer equations, hence the minimum depth of 10 m. The horisontal advection schemes for momentum and temperature are the default third-order upwind scheme that comes with ROMS, while in the vertical, default fourth order centered schemes are used. For salinity we use the HSIMT advection scheme in both the horisontal and the vertical. There is no explicit lateral eddy viscosity in the interior of the domain but we do use a lateral diffusivity for tracers of 10 $m^2s^{-1}$.

## 2.3 Turbulence mixing scheme

The vertical turbulent viscosity and diffusivities are obtained from the ROMS implementation of the Generic Length Scale (GLS) equations (Umlauf and Burchard, 2003), which are prognostic equations that solve for the turbulence kinetic energy, $k$ and a "generic length scale", $\psi$. This turbulence scheme contains a range of tunable parameters, and we use the default "gls" settings reported in Warner et al. (2005). When a perfect restart of the model cannot be made (e.g. following an analysis cycle),





the GLS equations are initialised using the minimum values for $k$ and $\psi$ provided in the input, and the numerical values are here taken to be $10^{-8}$ for both $k$ and $\psi$. Dedicated stability functions derived using second order closure relate the size of the eddy viscosity and diffusivities to the gradient Richardson number. In Norkyst we use the "CANUTO_A" option, which is based on the work of Canuto et al. (2001).

## 3 Forcing

### 3.1 Atmospheric forcing

For operational forecast production, Norkyst uses atmospheric forcing from the numerical weather prediction (NWP) system AROME-MetCoOp (Müller et al., 2017), which is part of the operational services of MET Norway. AROME-MetCoOp has a horisontal resolution of 2.5 km and a temporal output resolution of 1 h. The AROME-MetCoOp model domain does not fully overlap with the Norkyst domain—small parts of the south-west and north-west corners of the Norkyst grid are outside of the 95 AROME-MetCoOp grid—and we use data from the ECMWF (medium range HRES from IFS) to fill the gaps. Note that for the period 2012-16 AROME-MetCoOp was not operational, and for the hindcast archive mentioned earlier we used atmospheric data from a 3 km simulation of the Weather Research and Forecasting model (WRF), see Asplin et al. (2020) for details.

Norkyst uses NWP data for relative humidity and temperature evaluated at 2 m height, and NWP winds evaluated at 10 m height. Radiative fluxes are calculated using the COARE 3.0 bulk flux scheme that is part of ROMS. We supply shortwave 100 and downwelling longwave radiation fluxes directly from the NWP system, leaving the COARE algorithm to calculate the net longwave radiation fluxes. Note that the downwelling short and longwave fluxes for the years 2012-20 were retrieved from the regional atmospheric hindcast NORA3 (Solbrekke et al., 2021), while the same fluxes were provided by AROME-MetCoOp for the remaining years. The atmospheric pressure is also included in the forcing, hence storm surge and tide-surge interactions are represented in Norkyst.

Internal heating due to short wave radiation is modelled assuming a Jerlov water type III (equivalent to the internal ROMS parameter "JWTYPE" set to the value 5). Average values from Sentinel-3/OLCI observations are consistent with this choice. As already mentioned, extensive sea ice formation is not common in the coastal area covered by Norkyst, and hence the model does not contain any sea ice module or coupling to a separate sea ice model. In Norkyst we have instead activated an option that suppresses further cooling if the water is at the freezing points, emulating the presence of an ice cover (option 110 "LIMIT_STFLX_COOLING").

### 3.2 Riverine forcing

All rivers in Norkyst are specified with volume flux, temperature and a vertical shape of the outflow. The vertical shape of the outflow is kept constant in simulations, but has a Froude number dependency on the volume flux such that freshwater outflows from smaller rivers are more confined to the surface layers (Albretsen et al., 2011). The volume fluxes for all the 115 Norwegian rivers are based on daily measurements from the Norwegian Water Resources and Energy Directorate (NVE,





https://nve.no), where total runoffs for 69 Norwegian coastal regions are distributed to 1760 main rivers (247 in the current operational setup) according to their upstream precipitation area. Runoffs from Swedish, Danish and Scottish rivers are obtained from the European hydrological predictions for the environment (E-HYPE) model (Donnelly et al., 2016).

The salinity is set equal to the low value 1 for all discharges. Due to a lack of *insitu* river temperature data, we estimate
runoff temperature applying a Gaussian function (maximum temperature August 5th and standard deviation 50 days) limited by a minimum temperature of 2 °C and a maximum temperature reduced by increasing latitude and/or how far into a fjord the outlet is positioned. The latter is implemented to compensate for cold glacier runoffs in many of the deep fjords along the Norwegian coast.

### 3.3 Lateral boundary conditions

The Norkyst model setup needs lateral boundary conditions on all four boundaries. In the south and in the north, the model has boundaries in shallow shelf seas with significant differences in tidal range. For instance, in the Skagerrak the tidal range is typically less than half a metre, while in the Barents Sea the range can be upwards of 3 m in the easternmost part of the model domain. The western boundary is an extensive open boundary that lies partly in the deep ocean. The eastern boundary runs through the central Kattegat. This boundary was placed to avoid the complex straits between Denmark and Sweden,
while ensuring that the fluctuating Kattegat–Skagerrak front remains entirely within the model domain. The main prevailing influxes are Atlantic water through the western boundary, North Sea water through the southern boundary, and Baltic Sea water through the eastern boundary. Complex water mass transformation processes along the coast leads to the formation of the Norwegian Coastal Current (NCC, e.g., Christensen et al., 2018). In general, the NCC and the water masses of Atlantic origin exits the domain in the north, either flowing into the Barents Sea proper or towards the Arctic ocean following the shelf
break northwards.

We primarily use lateral boundary conditions from the Copernicus Marine Environment Monitoring Service (CMEMS) and the Arctic Monitoring Forecasting Centre (ARC MFC), with the exception of the southern part of the eastern boundary (in the Kattegat), where we use boundary conditions from the Baltic Monitoring Forecasting Centre (BAL MFC). The boundary conditions are implemented as in Marchesiello et al. (2001), using daily averages from the coarse resolution models, and
using ROMS to apply lateral tidal forcing with tidal constituents obtained from TPXO9 (Egbert and Erofeeva, 2002). We also use a relaxation zone that is 40 grid points wide, in which we increase the lateral eddy diffusivities gradually from 10 to 50 $m^2s^{-1}$ from the interior to the boundaries. For tracers and 3D velocities we apply a combined radiation/nudging scheme, with a nudging time scale of 15 days on outgoing signals, changing to 0.5 days for incoming signals. For the sea surface height and barotropic velocities, we use the "Chapman explicit" and "Shchepetkin" options, respectively. The external models
used to force the lateral boundaries do not contain the storm surge signal, hence we have activated the preprocessing option "PRESS_COMPENSATE" to apply the inverse barometer effect on sea surface height data.





## 4   Model evaluation

The hydrographic properties along the Norwegian coast in Norkyst are evaluated against the measurements from Institute of Marine Research (IMR)'s fixed coastal stations (see e.g., Albretsen et al., 2012, and available data from http://www.imr.no/
forskning/forskningsdata/stasjoner). In addition, we show a comparison with detailed current and hydrography measurements from the Hardangerfjord to illustrate the model performance in the fjords.

Temperature and salinity profiles are measured 2–4 times per month, presently with RBR CTD sondes (https://rbr-global. com). The position of the seven coastal stations from Lista in the south to Ingøy in the north are shown in Fig. 1. The hydrographic properties in the offshore open waters are evaluated against the gridded CORA Dataset of temperature and salinity
(Szekely et al., 2024). CORA is a global product of reprocessed in-situ measurements available through the CMEMS data portal. Spatial resolution is 0.5 degrees in longitude, and varying in latitude from 0.5 degrees at the equator to 0.2 degrees at the North Pole. The vertical resolution is also variable with 87 layers and higher resolution closer to the surface.

Current measurements were made at a location in the middle of the Hardangerfjord at N59 59.49, E05 54.87. This location is named Hardangerfjord East ("HfjE"). The bottom depth at this location is 540 m. The current mooring consists of two
profiling current meters positioned at approximately 40 m depth. These are one Nortek Signature 250 measuring downwards in 2 m vertical bins and one Nortek Aquadopp measuring upwards in 1 m vertical bins (https://www.nortekgroup.com). The instruments measure for 4 minutes every 20 minutes. Additional CTD measurements are from the location "H2" in the inner part of the Hardangerfjord at N60 23.32, E06 20.51. The bottom depth at this location is  850 m. Profiles were measured during monthly regular cruises. The instruments used are the SAIV SD204 (prior to 2019; http://www.saivas.no) and the RBR
Concerto 3 or Maestro 3 (https://rbr-global.com).

For the comparison, we use a hindcast archive starting from 2012, and which is being regularly updated. It should be noted that a continuous and internally consistent high resolution archive of atmospheric forcing data has not been available for the entire period. A warm bias reaching up to 0.7-0.8 degrees in the surface layer in summer, resulting from too high radiation forcing, has been identified when using NORA3 data (period 2012-2020, see Gonzalez et al., 2025), which is discussed further
below in Sec. 4.2.

### 4.1   Open ocean error statistics

In open water, the hindcast simulation is compared with the gridded in-situ measurements of temperature and salinity from the CORA dataset. The validation period spans from 2015 to 2022, with data categorized into four depth ranges: surface layer (0 to 20 meters), upper layer (20 to 100 meters), intermediate layer (100 to 250 meters), and deep layer (beyond 250 meters).
The CORA dataset covers the entire Norkyst domain, and we provide a validation for the entire area rather than focusing on specific regions within Norkyst. Temperature measurements are more prevalent than salinity measurements, with the majority of observations concentrated in the surface water (0 to 20 meters). Data availability is more limited in the deeper parts of the water column.





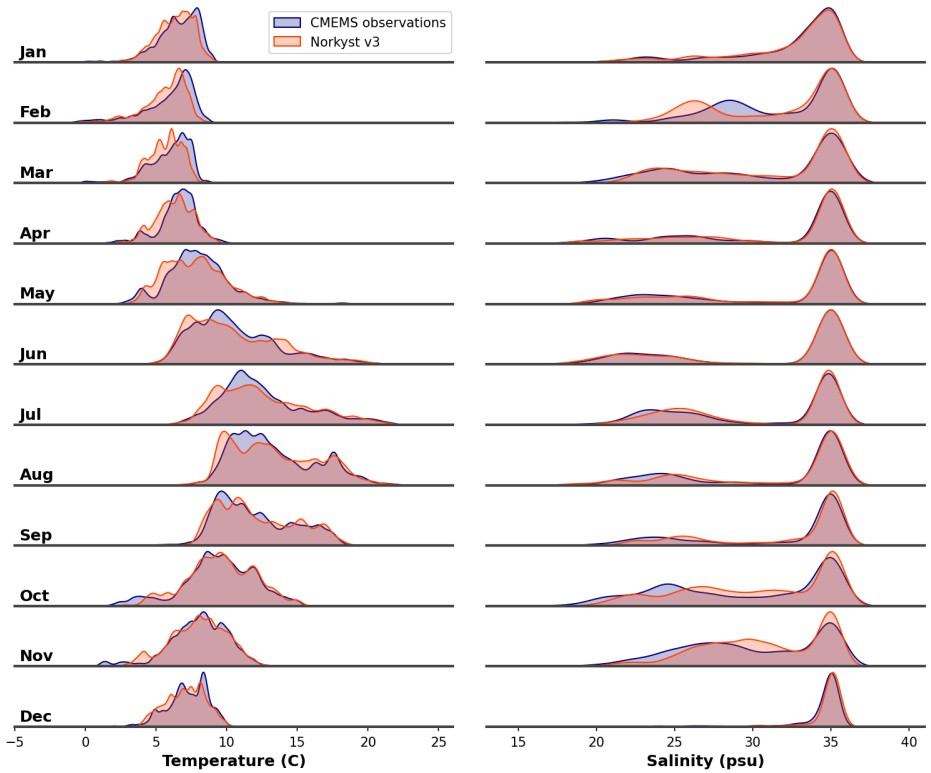

**Figure 2.** The figure shows a comparison of the temperature and salinity distributions in the upper 20 meters between the Norkyst v.3 hindcast archive and the gridded CORA dataset of in-situ measurements from Copernicus Marine Service. The comparison covers the years 2015 to 2022.

We find that the modeled salinity and temperature distributions in the upper 20 meters closely align with observed data, see Figure 2. This alignment indicates that the model is capturing environmental conditions and is accurately simulating the seasonal and spatial variations in salinity and temperature. While there may be differences from year to year, overall this gives us confidence in the capability of Norkyst to accurately predict hydrographic properties in offshore open areas.

Figures 3 to 6 present the monthly systematic error (bias) and the magnitude of error (RMSE) for temperature and salinity across various depths. In the surface layer (0 to 20 meters) and upper layer (20 to 100 meters), a small positive bias is observed during the first part of autumn, indicating that the model tends to be slightly warmer than actual measurements. Conversely, a negative bias during winter suggests the model is cooler than observed this part of the year, while the summer months are largely bias free. The intermediate layer (100 to 250 meters) consistently exhibits a cold bias, while the deep layer (beyond 250 meters) show a warm bias. These results represent mean values over the years 2015 to 2022. The magnitude of error generally remains within 1 degree Celsius across most depths and months, with errors being less pronounced in surface waters compared to deeper layers.





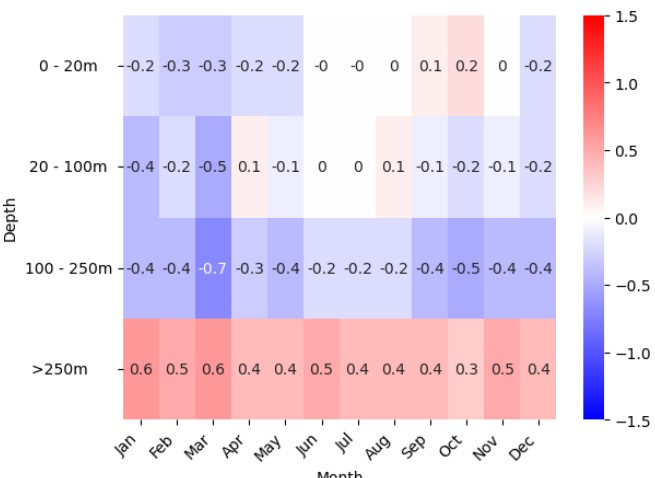

**Figure 3.** This heatmap illustrates the temperature bias between the Norkyst v3 hindcast and the CORA dataset, analyzed monthly from 2015 to 2022. The data is categorized by depth: the surface layer 0 to 20 meters, the upper layer covers 20 to 100 meters, the intermediate layer ranges from 100 to 250 meters, and the deep layer includes all depths beyond 250 meters.

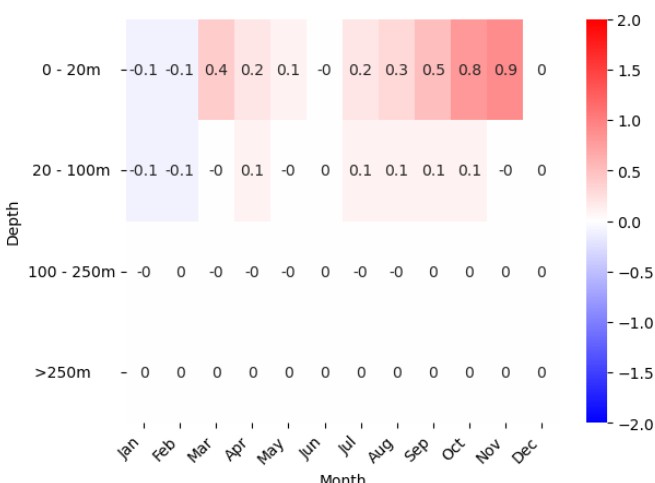

**Figure 4.** This heatmap illustrates the salinity bias between the Norkyst v3 hindcast and the CORA dataset, analyzed monthly from 2015 to 2022. The data is categorized by depth: the surface layer 0 to 20 meters, the upper layer covers 20 to 100 meters, the intermediate layer ranges from 100 to 250 meters, and the deep layer includes all depths beyond 250 meters.





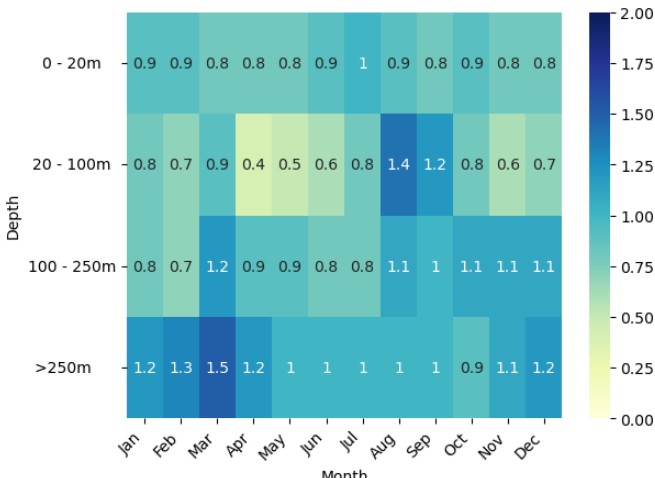

**Figure 5.** This heatmap illustrates the temperature RMSE between the Norkyst v3 hindcast and the CORA dataset, analyzed monthly from 2015 to 2022. The data is categorized by depth: the surface layer 0 to 20 meters, the upper layer covers 20 to 100 meters, the intermediate layer ranges from 100 to 250 meters, and the deep layer includes all depths beyond 250 meters.

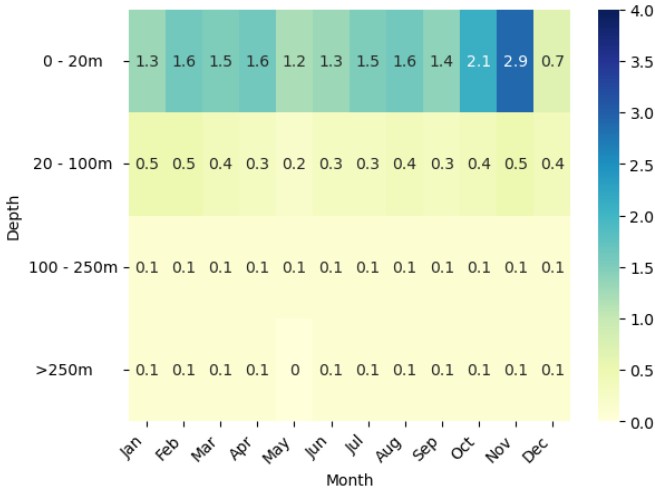

**Figure 6.** This heatmap illustrates the salinity RMSE between the Norkyst v3 hindcast and the CORA dataset, analyzed monthly from 2015 to 2022. The data is categorized by depth: the surface layer 0 to 20 meters, the upper layer covers 20 to 100 meters, the intermediate layer ranges from 100 to 250 meters, and the deep layer includes all depths beyond 250 meters.





Salinity observations are limited at depths beyond 100 meters, yet the available data indicates small errors at these levels. In contrast, at the surface, where observations are more abundant, the model typically shows a positive bias. It is not clear yet whether this bias is primarily due to the boundary conditions, the river discharge estimates, or internal processes in the model.

## 4.2   Coastal ocean error statistics

The hindcast archive has also been compared with the temperature and salinity profiles from the IMR fixed coastal stations for the period 2012-2023, and we have focused on the two vertical levels at 10 and 150 m depth, representing the surface layer and intermediate layer, respectively. Table 1 and 2 display relevant statistics from all the coastal stations for temperature and salinity, respectively. Besides observed and modelled averages and standard deviations, we have listed the systematic error (bias), the mean absolute error (MAE), the root mean square error (RMSE), the normalized error and the correlation coefficient.

The hydrographical properties in the surface layer fit well with the observations, with normalized biases below 0.24° C and 0.76 for temperature and salinity, respectively, for all stations. The model is slightly more biased at the northernmost stations, and we also find that correlation coefficients for salinity at Skrova and Ingøy are lower than for the other stations. A complete time series from the station Outer Utsira is displayed in Figure 7, which demonstrates that the model closely follow the observations in the southern part of the domain.

From validation of forcing data and near-surface temperatures in the Norkyst archive we have discovered that the solar radiation in the NORA3 archive is slightly exaggerated, most likely due to underestimated clouds. The Norkyst hindcast applied these fluxes for the years 2012-20. By rerunning 2021 with atmospheric fluxes from AROME MetCoOp only, we found that the NORA3 fluxes introduced a warm bias during summer of about 0.7-0.8° C compared with solar radiation provided by AROME MetCoOp. On shorter time and spatial scales, we can expect anomalously exaggerated water temperatures near the surface in the hindcast, and especially visible inshore.

The intermediate depth layer along the Norwegian coast is reproduced realistically. Table 1 shows that the 150 m temperatures have small biases and high correlation except for a normalized bias of -0.96° C and -1.12° C at the stations Sognesjøen and Skrova, respectively. Also, the correlation coefficient at the Skrova station is too low. Similarly, the 150 m salinities at Sognesjøen and Skrova are less well represented in Norkyst with relatively high bias and low correlation coefficient (Table 2). The latter low correlation is also seen in the intermediate layer salinities at Outer Utsira, Eggum and Ingøy. A complete time serie from the station Skrova is shown in Figure 8, and we see that while the model salinity is underestimated, the model temperature shows too high variability.

## 4.3   Comparison with observations from the Hardangerfjord

The location "HfjE" is situated about 60 km in from the Hardangerfjord mouth, and about one third into the fjord. The fjord
width is approximately 3.6 km here, which is typical for the general fjord width although the fjord has many bends, fjord arms and narrow passages. Currents have been measured for a long time at this location, and are found to characterize the lateral water exchange reasonably well. The along-fjord current component, rotated 45 degrees clockwise from the North, is varying episodically by the influence of different forcing mechanisms like the wind, tides, and horizontal internal pressure differences.





| | | Lista | Utsira | Sognesjøen | Bud | Skrova | Eggum | Ingøy |
|---|---|---|---|---|---|---|---|---|
| Temperature | Mean obs | 10.59 | 10.63 | 10.66 | 9.79 | 7.85 | 8.20 | 7.01 |
| 10 m depth | Mean mod | 10.40 | 10.84 | 10.52 | 9.81 | 8.66 | 8.62 | 7.38 |
| | Std. dev. obs | 4.23 | 3.70 | 3.20 | 3.13 | 3.40 | 2.62 | 1.93 |
| | Std. dev. mod | 4.42 | 3.87 | 3.79 | 3.21 | 3.28 | 2.72 | 2.21 |
| | Bias | 0.07 | 0.14 | 0.19 | 0.17 | 0.72 | 0.48 | 0.46 |
| | MAE | 0.67 | 0.61 | 0.58 | 0.39 | 0.81 | 0.57 | 0.60 |
| | RMSE | 0.97 | 0.95 | 0.77 | 0.52 | 1.02 | 0.80 | 0.81 |
| | Norm. bias | 0.02 | 0.04 | 0.06 | 0.06 | 0.21 | 0.18 | 0.24 |
| | Corr. coef. | 0.98 | 0.97 | 0.98 | 0.99 | 0.98 | 0.97 | 0.96 |
| | | | | | | | | |
| Temperature | Mean obs | 7.81 | 7.92 | 8.07 | 8.18 | 7.32 | 7.24 | 6.51 |
| 150 m depth | Mean mod | 7.63 | 7.66 | 7.60 | 7.73 | 6.99 | 6.81 | 6.25 |
| | Std. dev. obs | 0.96 | 0.78 | 0.38 | 0.62 | 0.34 | 0.74 | 1.03 |
| | Std. dev. mod | 0.92 | 0.71 | 0.60 | 0.78 | 0.89 | 0.74 | 1.09 |
| | Bias | -0.10 | -0.15 | -0.36 | -0.39 | -0.38 | -0.46 | -0.38 |
| | MAE | 0.36 | 0.26 | 0.42 | 0.50 | 0.84 | 0.51 | 0.48 |
| | RMSE | 0.50 | 0.42 | 0.49 | 0.58 | 0.98 | 0.60 | 0.59 |
| | Norm. bias | -0.11 | -0.19 | -0.96 | -0.63 | -1.12 | -0.62 | -0.37 |
| | Corr. coef. | 0.86 | 0.87 | 0.85 | 0.84 | 0.16 | 0.87 | 0.91 |

**Table 1.** Temperature statistics from measurements and the Norkyst model from the seven fixed, coastal hydrographical stations along the Norwegian coast, using the period 2012-2023. First column denotes parameter and depth, next column list the different statistical elements while the measures are displayed for the positions off Lista (southern tip of Norway), Utsira, Sognesjøen and Bud (Norwegian west coast) and Skrova, Eggum and Ingøy (northern Norway). The mean values and standard deviations are based on full time series, i.e., daily averages from the model, while the biases (model - observation), mean absolute errors (MAE), root mean square errors (RMSE), normalized biases (bias divided by the observed standard deviation) and the correlation coefficients are based on all values where both measurement and model estimate exist. The numbers in red indicate where we consider that there is greatest potential for improvement in the model system.



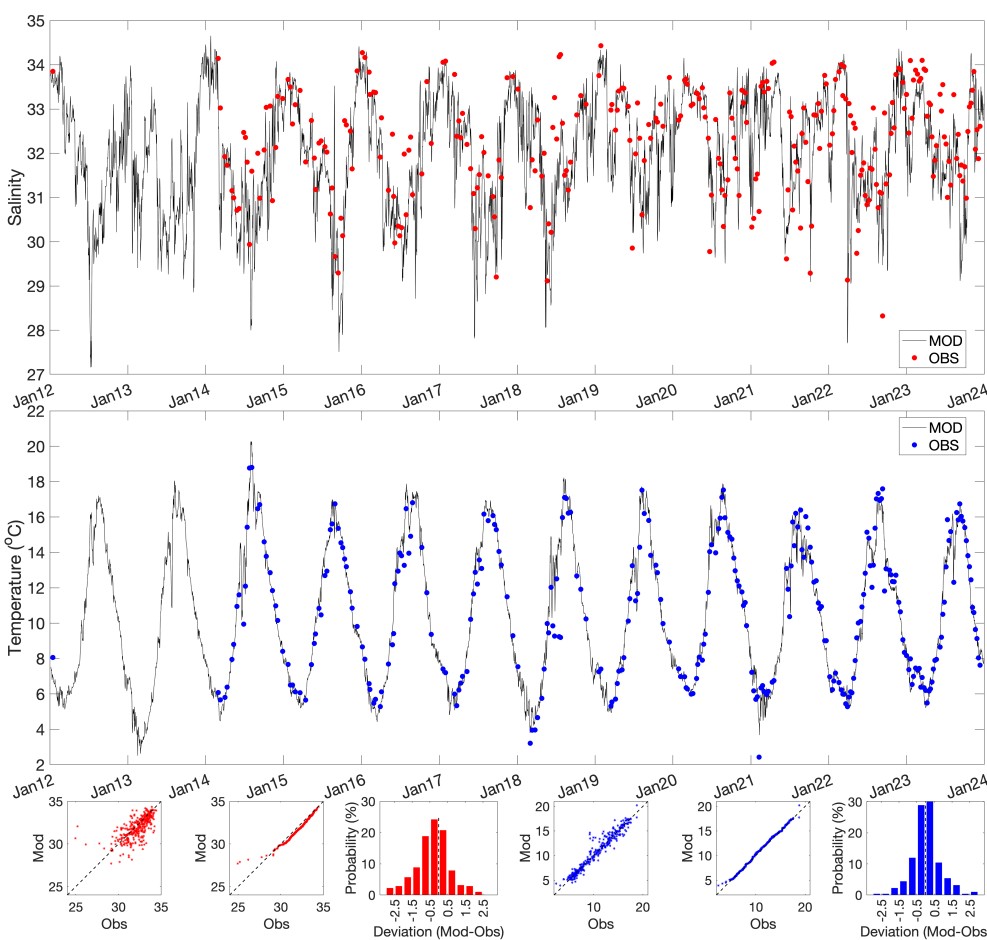

**Figure 7.** Salinity (top panel) and temperature (mid panel) values from the fixed station Outer Utsira at 10 m depth from 2012 to 2023. Model values are shown as black lines while observed salinity and temperature are denoted with red and blue dots, respectively. The lowermost panels summarize scatter plots, qq plots (percentiles) and histogram of deviations for both salinity (red) and temperature (blue).



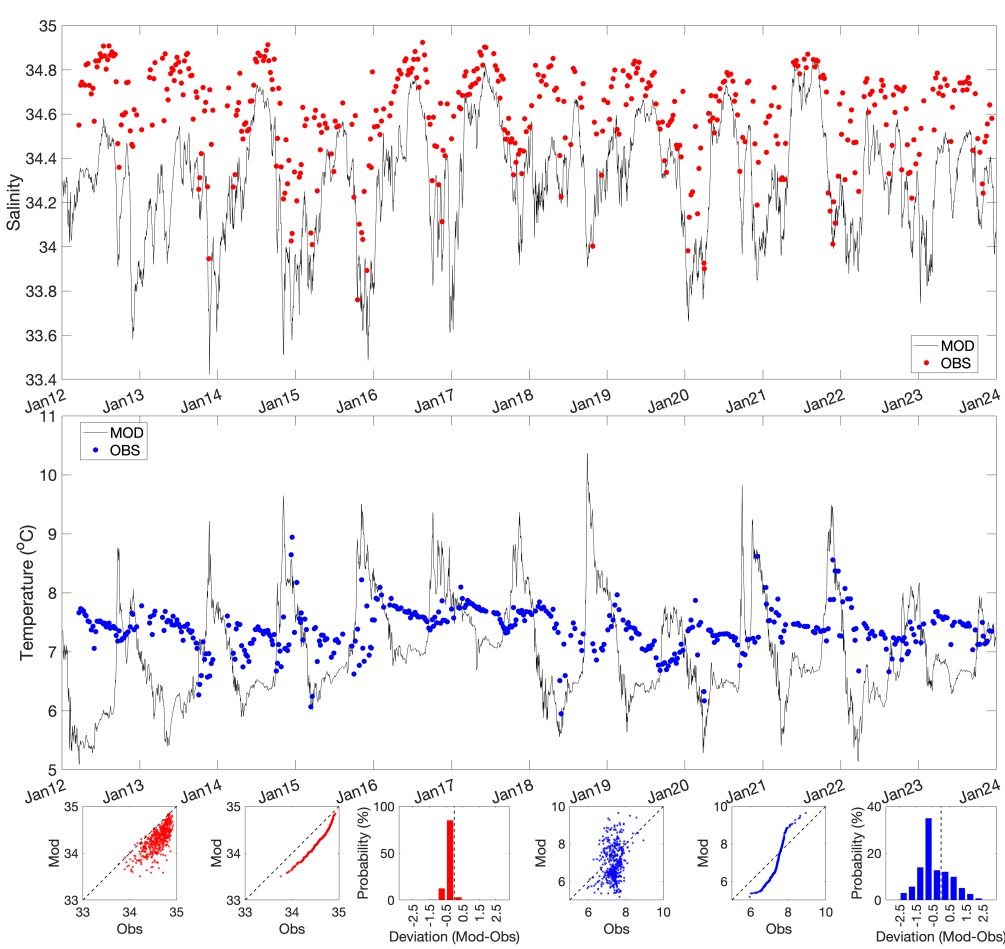

**Figure 8.** Same as Figure 7 but from the fixed station Skrova and from 150 m depth.



|  |  | Lista | Utsira | Sognesjøen | Bud | Skrova | Eggum | Ingøy |
|---|---|---|---|---|---|---|---|---|
| Salinity | Mean obs | 31.61 | 32.43 | 31.74 | 32.83 | 32.73 | 33.61 | 34.16 |
| 10 m depth | Mean mod | 31.09 | 31.92 | 31.20 | 32.91 | 33.14 | 33.72 | 34.08 |
|  | Std. dev. obs | 2.26 | 1.16 | 1.25 | 0.82 | 0.53 | 0.32 | 0.21 |
|  | Std. dev. mod | 1.96 | 1.31 | 1.34 | 0.75 | 0.44 | 0.38 | 0.26 |
|  | Bias | -0.47 | -0.36 | -0.36 | 0.12 | 0.40 | 0.13 | -0.03 |
|  | MAE | 1.09 | 0.76 | 0.68 | 0.33 | 0.49 | 0.24 | 0.19 |
|  | RMSE | 1.49 | 1.03 | 0.87 | 0.45 | 0.58 | 0.30 | 0.25 |
|  | Norm. bias | -0.21 | -0.31 | -0.29 | 0.15 | 0.76 | 0.40 | -0.16 |
|  | Corr. coef. | 0.77 | 0.70 | 0.81 | 0.86 | 0.61 | 0.72 | 0.50 |
|  |  |  |  |  |  |  |  |  |
| Salinity | Mean obs | 34.90 | 34.97 | 34.95 | 34.76 | 34.58 | 34.66 | 34.66 |
| 150 m depth | Mean mod | 34.90 | 34.95 | 34.86 | 34.66 | 34.31 | 34.55 | 34.52 |
|  | Std. dev. obs | 0.17 | 0.12 | 0.07 | 0.23 | 0.20 | 0.18 | 0.16 |
|  | Std. dev. mod | 0.24 | 0.14 | 0.17 | 0.28 | 0.26 | 0.18 | 0.19 |
|  | Bias | -0.01 | -0.01 | -0.09 | -0.09 | -0.29 | -0.12 | -0.14 |
|  | MAE | 0.13 | 0.10 | 0.13 | 0.16 | 0.30 | 0.18 | 0.19 |
|  | RMSE | 0.17 | 0.16 | 0.17 | 0.21 | 0.35 | 0.22 | 0.25 |
|  | Norm. bias | -0.05 | -0.09 | -1.29 | -0.41 | -1.45 | -0.64 | -0.88 |
|  | Corr. coef. | 0.69 | 0.37 | 0.54 | 0.73 | 0.72 | 0.43 | 0.38 |

**Table 2.** Same as Table 1, but for salinity.

At 10 m depth the maximum inflow is more than 0.6 ms$^{-1}$ and maximum outflow around 0.5 ms$^{-1}$ (Fig. 9). Deeper down,
the current strength weakens, but at 30 m depth most of the episodic in and outflows are present. At 100 m depth the absolute
current velocity is less than 0.2 ms$^{-1}$ and here the flow is less affected by the upper layer events. The tidal flow is also more
visible at this depth, and at this relatively deep and wide part of the fjord the tide oscillates with an amplitude of approximately
0.05 ms$^{-1}$. While comparing the observed along-fjord current with the results from Norkyst, we characterize the fit in three
classes: Good, medium and bad following Dalsøren et al. (2020). This method shows that the model results are good for 78%,
75% and 90% of the time at 10, 30 and 100 m depths respectively (Fig. 9).

The location H2 is situated about 120 km into the Hardangerfjord from the coast. At this location a seasonal surface brackish
layer of 5-10 m thickness exists. The temperature here vary seasonally between about 4° C during winter and about 18° C
during summer at the surface. Further down the seasonal variability is less and at 100 m depth the variability is only 2-3° C
centered at 8-9° C. The warmest period at depth is now shifted 6 months compared to the surface when winter is warmest and
summer coldest (Figure H2). The salinity at H2 has a seasonal variation too with fresher water of salinity less than 5 at 3 m
depth during the summer and >30 during winter (Fig. 10). At 30 m depth the variability of salinity is only within 31 to 35 and



**Figure 9.** Time series of along-fjord current from observations and model results at 10 m, 30 m and 100 m depths at the location Hardanger-fjord East from January to April 2024. The categories "good", "medium", "bad" are here defined as in Dalsøren et al. (2020), with category "good" implying correct direction of the flow and low bias, "medium" correct direction but high bias, and "bad" when the flow is in the wrong direction.

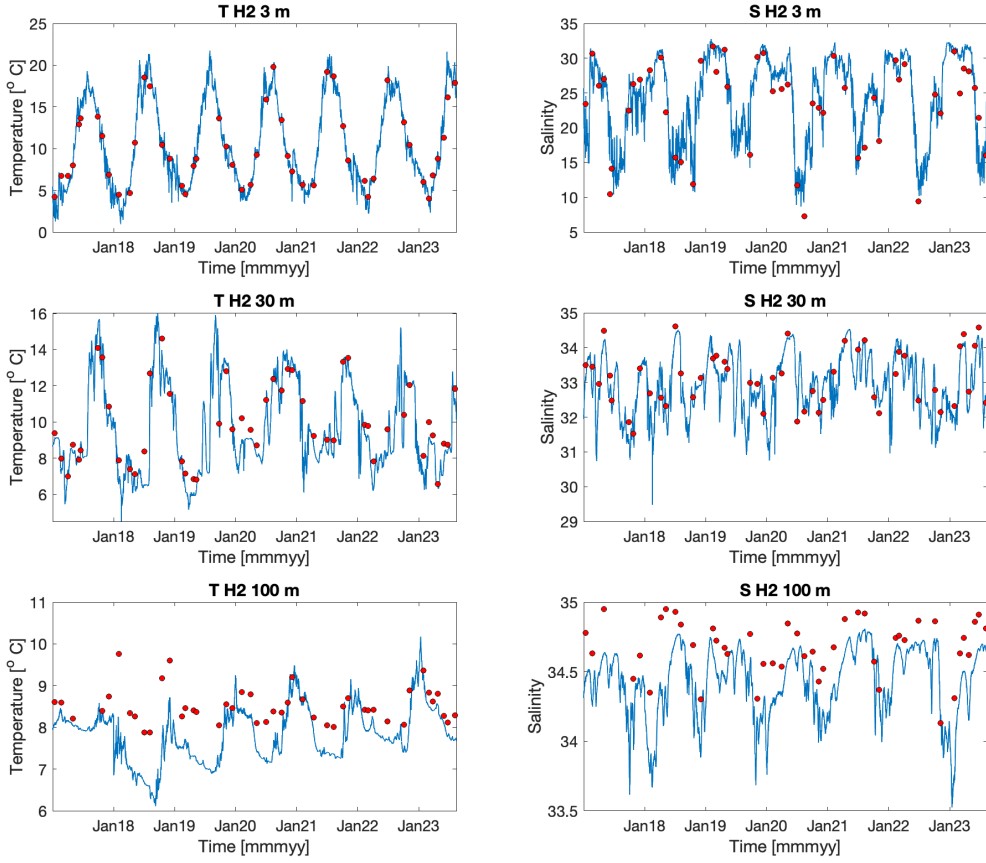

**Figure 10.** Time series of temperature (left) and salinity (right) from 3 m, 30m and 100 m depths at the location H2 in the Hardangerfjord from 2017 through 2023. The observations are marked by red dots and daily mean values from Norkyst by solid blue lines.

at 100 m depth even less, only between 34 and 35. The numerical model capture the observed hydrography reasonably well at 3 m depth, with deviations typically less than one unit (Fig. 10). At 100 m depth there is a bias of the model results towards slightly colder and fresher water.

## 5  Operational implementation

The daily forecasts of Norkyst are produced as a part of the operational scheduling system at MET Norway. This is an auto-mated system which handles the production and processing of forecasts from many other weather, ocean, sea ice, and wave models, with strict requirements for operational robustness. In the daily operational cycle, the latest available input forcing fields are downloaded and preprocessed before the production of model forecasts begins.



## 5.1 Scheduling

Presently, the operational implementation of Norkyst is scheduled to run once per day. The daily forecasts are produced on MET Norway's high performance supercomputer facilities. Before the simulations begin, updated boundary data, climatological nudging fields, and atmospheric forcing are downloaded and interpolated to the Norkyst grid. Daily updated river runoff data from NVE's total discharge estimates for Norwegian rivers is downloaded and distributed to the river outlets in the model for the time up to the initial time. For the forecast period, the river transport is considered to be constant. The system produces 120h forecasts starting at 00UTC on the present day.

## 5.2 Data dissemination

For the daily operational forecasts, the results from the 3D variables are interpolated from the 40 terrain-following model s-layers to 15 fixed z-layers at depths = 0, 1, 2, 3, 5, 7, 10, 15, 25, 50, 65, 75, 100, 200 and 300 m. The hindcast data are available both on the native model coordinates and interpolated to 25 fixed z-layers at depths = 0, 1, 2, 3, 5, 7, 10, 15, 25, 50, 65, 75, 100, 200, 300, 400, 500, 750, 1000, 1250, 1500, 1750, 2000, 2250 and 2500 m. Output variables are the current components ("u_eastward", "v_northward" and "w"), "salinity", "temperature", and the sea surface elevation "zeta", as well as the vertical tracer diffusivity "AKs", and the wind components "Uwind_eastward" and "Vwind_northward". Links to the data are provided in the data availability statement later on.

## 6  Concluding remarks

The Norkyst coastal ocean circulation models is now in version 3. Compared with version 2, the main differences are an extended domain to cover the energetic flows along the Norwegian shelf break, better numerics, and improved forcing. Norkyst compares favourably with observations and does not display large biases. The horizontal resolution (800 m) is sufficient to resolve mesoscale eddies, and Norkyst faithfully reproduces the main dynamical features of the Norwegian coastal ocean circulation. The revised domain avoids cutting across the most eddy-active regions, and reduced hydrographic biases further lessen topographic constraints on the coastal current; both factors allow freer eddy shedding and influence conditions nearshore and offshore.

Hindcast and forecast data from Norkyst are freely available both as direct and post-processed (constant depth levels) outputs from ROMS, and also through APIs and various dissemination portals, see the data availability statement below. The modeling system is under continuous development, and ongoing efforts focus on the following activities:

– Data assimilation: A version of Norkyst with coarser resolution (2.4 km, "Norkyst-DA") with 4D-Var data assimilation has been in operational production since 2017. In ongoing work we are adapting the data assimilation scheme for v.3 of Norkyst (the current assimilative setup is described e.g. here: Iversen et al., 2023), investigating a mixed resolution, mixed precision approach, with reduced resolution and precision in the inner loops.



– Ensemble prediction: As Norkyst is part of various decision support systems, it is important to provide information about forecast uncertainty to downstream users. The operational implementation of Norkyst is therefore configured for parallel execution of several model instances for the purpose of ensemble prediction.

    – Two-way nested fjord models: ROMS has functionality for two-way nesting between model grids of different resolution, allowing exchange of data across grids at time step level. This functionality is used for very high resolution fjord models

using a 5:1 refinement (i.e., 160 m horisontal resolution) in selected areas, providing a seamless description of the physical state of the ocean all the way from the innermost part of the fjords out to the open ocean. At present we have two-way nested fjord models for the Oslofjord region and one region of Western Norway (Mørekysten), with more systems being added in the future.

    – Forcing: Coupling to surface waves through the virtual wave stress and Coriolis-Stokes force, in addition to injection of

turbulence kinetic energy associated with wave dissipation, has been tested for various applications (e.g., Röhrs et al., 2014). Operational wave prediction systems already use Norkyst data to model refraction by currents, enabling weak two-way coupling between the mean circulation and the waves. In addition, the implementation of the lateral boundary conditions is also under investigation, and planned activities include development of boundary data bias correction schemes.

– Sea ice: The Norwegian Coast is largely ice free, and Norkyst v.3 does not contain a sea ice component. Occasionally some fjords will partially freeze over during cold spells in winter, posing a challenge to maritime operations. Work is underway to investigate the applicability of a sea ice model that has recently been integrated with the flavour of ROMS we are using.

*Code and data availability.* A continuously aggregated archive of the daily forecasts at hourly temporal resolution is available at https://
thredds.met.no/thredds/fou-hi/norkystv3.html. The hindcast archive is available at https://thredds.met.no/thredds/catalog/romshindcast/norkyst_v3/catalog.html. Some fields can also be accessed from the API https://api.met.no/. Zenodo contains copies of the Norkyst ROMS configuration (Christensen et al., 2025b), a copy of the full ROMS model code (Rutgers, The State University of New Jersey, 2021), and the analysis and processing scripts used to produce the results presented here (Christensen et al., 2025a).

*Author contributions.* KHC coordinated the work and contributed to model developments and writing; JA, LA, ADS, YG, HGF, MS, AKS
and MT contributed with model developments, verification, and writing; SCI, MFJ, IAJ, PNS, JS and NMK contributed with model developments and verification.

*Competing interests.* No competing interests are present.



*Acknowledgements.* The development of Norkyst has been supported by core funding from MET Norway and the Institute of Marine Research, and in addition by the Research Council of Norway through the projects "KnowSandeel 3.0" (Grant agreement 352755), "SFI Blues" (Grant agreement 309281); and the EU Commission through the project "Forecasting and observing the open-to-coastal ocean for Copernicus users" (FOCCUS, Grant agreement 101133911). The Norkyst topography is partly based on EMODnet Bathymetry Consortium (2020): EMODnet Digital Bathymetry (DTM), see https://doi.org/10.12770/bb6a87dd-e579-4036-abe1-e649cea9881a. The historical simulations of Norkyst were performed on resources provided by Sigma2—the National Infrastructure for High-Performance Computing and Data Storage in Norway.



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
