# Peer review of ""Norkyst" version 3: the coastal ocean forecasting system for Norway"

_EGUsphere, 2025_

## Referee Comment (RC2)

**"Norkyst" version 3: the coastal ocean forecasting system for Norway**

Kai Håkon Christensen1,3, Jon Albretsen2, Lars Asplin2, Håvard Guldbrandsen Frøysa2, Yvonne Gusdal1, Silje Christine Iversen1, Mari Fjalstad Jensen2, Ingrid Askeland Johnsen2, Nils Melsom Kristensen1, Pål Næverlid Sævik2, Anne Dagrun Sandvik2, Magne Simonsen1, Jofrid Skarðhamar2, Ann Kristin Sperrevik1, and Marta Trodahl1,†

Correspondence: Kai H. Christensen (kaihc@met.no)

**Abstract.** We describe the operational forecasting system "Norkyst", now in version 3, which is used for predicting the ocean circulation along the coast of mainland Norway and in the fjords. The forecasting system is based on the Regional Ocean Modeling System (ROMS), and has sub-kilometric horisontal resolution to resolve mesoscale features. Here we describe the basic configuration and report verification statistics of unconstrained model runs. The main features of the circulation and hydrography, including seasonal variations, are well represented in Norkyst v.3, making the forecast system suitable for its intended use as an open service for users in public or private sectors such as aquaculture, fishery, shipping, research, consulting, environmental management, and others who needs detailed predictions of the physical state of the Norwegian coastal ocean.

**1 Introduction**

Mainland Norway has a very long coastline which stretches from the Skagerrak in the south to the Barents Sea in the north (Fig. 1). As the name implies, the Norwegian coast is a gateway to the Arctic. The main ocean currents in this region are (i) the Norwegian Atlantic Current (NwAC), an extension of the North Atlantic Current that carries warm, saline water to the Arctic Ocean, and (ii) the Norwegian Coastal Current (NCC), originating in the Skagerrak from mixed waters of the North Atlantic, North Sea, Baltic Sea, and river run-off, flowing along Norway's coast into the Barents Sea (e.g., Sætre, 2007). The coastline itself is rather complex, with many long and deep fjords and more than 200.000 large and small islands. The coastal waters are often strongly stratified and the shelf circulation is dominated by energetic eddies.

There are large variations in the prevailing weather conditions along the coast. The coastal climate in the south-eastern parts of Norway is temperate while it is boreal in the north. Seasonal cycles in radiative forcing also have large amplitudes, with about one third of Norway being north of the polar circle. The day length variations cause the pronounced spring blooms that inspired Sverdrup (1953) to develop his "critical depth theory", continuing a long tradition of scientific investigations, starting with Helland-Hansen and Nansen (1909), linking the ocean physics to its biology to better understand the behaviour of the large

<sup>1Norwegian Meteorological Institute, Oslo, Norway

<sup>2Institute of Marine Research, Bergen, Norway

<sup>3University of Oslo, Oslo, Norway

<sup>†Now at Equinor, Stavanger, Norway

[revised manuscript text omitted]

---

## Author Comment (AC1)

I am not writing a full review, but I have just a three comment to the issues which caught my attention.

Lines 60-71, Sec. 2.1 Model grid: "The horizontal resolution of Norkyst is approximately 800 m. ROMS uses C grid staggering, and the domain size is 2747 x 1148 horizontal grid points. .... About 42% of the grid is land....", also Figure 1:

This is, indeed, an inefficient way to set up a grid.

The goal of this project, as I understand it, is to make an operational model, which both ambitious and computationally challenging, as it involves a large, high-resolution grid, obviously very expensive to run, and even to handle the associated data. And is intended to be used operationally on daily basis.

It happened, that Norway has the most complicated coastline in the World, almost fractal, and going inside these fjords is what is desired, but the grid must be fine enough to even get crude idea about what is going on there. So the authors choose to setup a simple rectangular grid, covering the entire country, and, obviously, facing the need to make a compromise between grid resolution and how far offshore you can go. Cutting Lofoten basin in half also means that you are cutting eddies in half. Lofoten basin is famous for its eddies.

My suggestion: make it curvilinear:

http://91.225.115.25/norkyst/v6.1/roms_grid.pdf

The grid is designed to cover larger area, but in such a way that resolution contracts when approaching Norwegian coast - kind of alternative to 2-way nesting, except that it has no steps and is perfectly smooth.

In this variant the dimensions are 3844 x 802, which slightly less than the total number of points of your grid 2747 x 1148. Note that only one out of several grid lines is shown (it is not practical to plot them all), but the land mask is the actual land mask as it would be seen by ROMS code,

running using this grid.

Its grid resolution is shown here

http://91.225.115.25/norkyst/v6.1/roms_grid_res.pdf

By the design the grid has the same resolution in both directions, locally pm=pn at any point, even though bothh of them change by more than order of magnitude, depending on the location. 500 means 500 meters; 1k2 means 1.2km.

Topography looks like this:

http://91.225.115.25/norkyst/v6.1/roms_grid_topo.pdf

There are three variants of this grid, which differ only by resolution:

http://91.225.115.25/norkyst/v6.1/    xi_rho=3844 eta_rho=802
http://91.225.115.25/norkyst/v6.2/    xi_rho=4611 eta_rho=962
http://91.225.115.25/norkyst/v6.3/    xi_rho=3076 eta_rho=642

The content of these directories is as follow:

"roms_grid.nc" -- netCDF file for ROMS grid, which can be made runnable.  This file already contains all finalized variables defining grid as geometric object (lon_rho,lat_rho, lon_psi,lat_psi, pm,pn,angle,f) and place-holders for land mask and topography (currently "mask_rho" is generated from USGS GSHHS data. Alternatively EMODNET Europe_coastline_2020_OSM.shp can be used. No effort was made thus far to do any manual editing of land mask, so some narrow passages should be inspected and opened as needed), and "hraw" interpolated/averaged from GEBCO_2024.nc.

All other files are precursors, configuration files, diagnostics, orthogonality check, etc...

Many thanks, this is a very generous effort to improve our grid! The forecasting system is already operational, producing daily forecasts since 2012, and we have admittedly been quite conservative when updating our grid for this version 3. Our strategy so far has been to use the nesting options to improve the resolution where needed, and we have also been thinking about using nesting options to reduce the percentage of land points. There are some open questions about forcing data and ability to resolve some aspects of the open ocean dynamics that we would need to look into, but you have certainly given us something to think about for version 4.

Lines 73-76, "The main challenge with regards to stability is not associated with horizontal resolution, but with occasional large vertical velocities in regions of strong convergence (e.g. at the Kattegat-Skagerrak front), which in turn leads to violations of the CFL criterion in the vertical tracer equations, hence the minimum depth of 10 m"

This problem has been solved 10 years ago:

https://www.sciencedirect.com/science/article/pii/S1463500315000530

Indeed, once the horizontal resolution becomes finer and finer, at certain point vertical advection becomes the most restrictive factor.  Investigation of where and how exactly model blows-up due to computational instability reveals that vertical Courant number is generally very small everywhere, except just in few places, "hot spots" - literally, a handful of grid points out of one million or so holds the simulation hostage.    What is going on there is some physical process resulting in something highly non-equilibrium. It may be a front with strong vertical mixing next to no mixing. A supercritical flow (horizontal advection velocity exceeds internal wave speed) becoming subcritical, resulting in violent generation of internal waves (similar to . An internal wave has nowhere to go because of the bottom raise,

resulting in growth of amplitude, and eventual shoaling and crashing. Back-and-forth tidal movements over a topographic ridge causing

generation of internal waves of supercritical (too steep topography) nature.

Whatever.

This is a very accurate problem description for our case. We just realised that your fix is in line for the next development version of ROMS and we are very much looking forward to testing it.

Lines 142-144, <<For the sea surface height and barotropic velocities, we use the "Chapman explicit" and "Shchepetkin" options, respectively...>> -- perhaps the proper way to call it "Riemann boundary conditions", or something of this sort, see Sec. 2.1.2 in

https://www.sciencedirect.com/science/article/pii/S146350031000082X

what is relatively new here, is that all Riemann solvers (or numerical methods relying on the idea of propagating something along characteristics) always use non-staggered grids: this makes sense, because Riemann invariants are linear combinations of prognostic variables, and it is natural to make them co-located on the grid. However, ROMS uses staggered grid, and there is no way around it.  So this needs to be dealt with, and Sec. 2.1.2 is about this. Another point to make is that one cannot separate b.cs. for free surface and for barotropic velocities: they are part of the same algorithm - the only reason why the discrete value of free surface needs to be "radiated-out" is to end up at the same point in space and time as the normal velocity component, so the two can be added/subtracted to form Riemann invariant.  Free surface value needs to travel only half-grid interval in space and one time step up in time, so if traced back to time step "n", the characteristic for free surface may be outside the grid interval (see Fig. 2 from Mason et al 2010). As the result, a naive (say "Chapman explicit") scheme causes more restrictive limitation on barotropic time step than the time stepping algorithm of barotropic mode.

Many thanks for these insights. We find that this treatment of the boundaries gives us much less artifacts compared to using the old "Flather" and "Chapman" conditions. We did have to spend some time tuning in the nudging time scales too, of course, and also have to match the exterior bathymetry in the relaxation zone.

---

## Author Comment (AC2)

**RC1**: 'Comment on egusphere-2025-3986', Isabel Garcia Hermosa, 07 Nov 2025

Dear authors, I am unsure of the tone of other articles in this journal. I find this article interesting, clear and pedagogical.

We thank the reviewer for all the constructive suggestions and positive feedback.

**Specific comments**

Regarding my comments below they are suggestions and questions.

I don't ask to do any changes regarding to this point, I would like to comment about it. I don't know if I understood correctly, but you say for the bathymetry you have a 10 m minimum depth criteria. With so many islands and channels isn't this a bit risky? Maybe I didn't understand clearly. I guess, as the other reviewer (Shchepetkin) mentions, it cold be due to the breach of the courant criteria. Perhaps there are other methods, but really dependent on the bathymetry available. Also if in general you get good results, it is ok. However coastal and shallow areas, with channels and shoals are very dependent on bathymetry.

Answer: We are confident that the results from our Norkyst model system with ROMS as the numerical core are realistic and not particularly affected by our choice of minimum depth as 10m. You are right that the Norwegian coast is extremely complex with small islands, narrow straits and fjord arms, and our horizontal resolution of approx. 800m x 800m will not allow resolving this coast line by any perfection. Our choice of minimum depth of 10m is connected to the CFL-criteria, but it holds well in most regions as the actual areas with depth less than 10m are close to land with very little impact on the main dynamics. We maintain higher resolution models as well (not described in this manuscript), and we then aim to better resolve both coast line and bathymetry when spatial resolution is increased.

I don't know sufficiently the area but cold the issues present in some of the stations in Tables and figures be an artifact of this depth minimum?

Answer: The validation statistics describe hydrographic quantities mainly, and are also from relative open (exposed) locations. Since most of the Norwegian coastal zone is stratified with a relatively shallow surface layer all year, and that the horizontal scales of salinity and temperature are large, we are very confident that the model biases are not a result of the chosen minimum depth.

We have added a sentence in line 64 (chapter 2.1, after we state the minimum depth):

"Our choice of minimum depth is based on a compromise between maintaining the CFL criterion (Courant-Friedrichs-Lewy) and the fact that most of the Norwegian coast is deeper than 10m. Exceptions are mainly found along the shoreline with negligible impact on the main hydrodynamics."

A suggestion regarding Section 4. For clarity, I would reorganise it or add a sub section (observations used in evaluation) after an introductory sentence on what you are comparing. Introduce the section with a sentence to make it clear (as mentioned early in the introduction, as a reminder) that this section compares the hindcast that has such and such characteristics to in situ observations. Then put the rest of the information in line 166 (period available for the hindcast).

Answer: We agree that we can increase the clarity in this section, and we have added a new sentence in the beginning of section 4 and altered the rest of the text between line 148-165:

4. Model evaluation

We present verification from the long hindcast, only, which is the unconstrained model version of the forecast system where we evaluate hydrographical properties in offshore and coastal waters and hydrodynamical properties in one of the largest Norwegian fjords.

The hydrographic properties in the offshore open waters are evaluated for 2015-22 against the gridded CORA Dataset of temperature and salinity (E.U. Copernicus Marine Service Information (CMEMS) 2024; https://doi.org/10.17882/46219). CORA is a global CMEMS product of reprocessed in-situ measurements, developed by the In Situ Thematic Assembly Center (INS-TAC)l. Spatial resolution is 0.5 degrees in longitude, and varying in latitude from 0.5 degrees at the equator to 0.2 degrees at the North Pole. The vertical resolution is also variable with 87 layers and higher resolution closer to the surface.

The hydrographic properties along the Norwegian coast in Norkyst are evaluated for the entire model period (2012-23) against the measurements from Institute of Marine Research (IMR)'s fixed coastal stations (see e.g., Albretsen et al., 2012, and available data from http://www.imr.no/forskning/forskningsdata/stasjoner). Temperature and salinity profiles are measured 2–4 times per month, presently with RBR CTD sondes (https://rbr-global.com). The position of the seven coastal stations from Lista in the south to Ingøy in the north are shown in Fig. 1.

From one of the largest Norwegian fjords, Hardangerfjord, the IMR has conducted current measurements at a location in the middle of the fjord at N59 59.49, E05 54.87. This location is named Hardangerfjord East ("HfjE"). The bottom depth at this location is 540 m. The current mooring consists of two profiling current meters positioned at approximately 40 m depth. These are one Nortek Signature 250 measuring downwards in 2 m vertical bins and one Nortek Aquadopp measuring upwards in 1 m vertical bins (https://www.nortekgroup.com). The instruments measure for 4 minutes every 20 minutes, and in our evaluation we show results from Jan-Apr 2024 only. We also apply additional CTD measurements for the period 2017-23 from the location "H2" in the inner part of the Hardangerfjord at N60 23.32, E06 20.51. The bottom depth at this location is 850 m. Profiles were measured during monthly regular cruises. The instruments used are the SAIV SD204 (prior to 2019; http://www.saivas.no) and the RBR Concerto 3 or Maestro 3 (https://rbr-global.com).

The following sentence (starting in line 166) looks out of place as it mixes things and the link is not clear to me. Perhaps add some text to clarify why it is there? I am assuming it is something like this below, I don't know if I understood correctly.

"It should be noted that a continuous and internally consistent high resolution archive of atmospheric forcing data TO BE USED IN THIS HINDCAST has not been available for the entire period. PLEASE NOTE THAT BECAUSE OF XXX, IN THE HINDCAST a warm bias reaching up to 0.7-0.8 degrees IS PRESENT in the surface layer in summer, resulting from too high radiation forcing. THIS has been identified when using NORA3 data (period 2012-2020, see Gonzalez et al., 2025), which is discussed further below in Sec. 4.2".

Answer: Thank you for this comment. We agree and have tried to elaborate and make it more clear by rewriting line 166-170:

It should be noted that a continuous and internally consistent high resolution archive of atmospheric forcing data to be applied in this hindcast has not been available for the entire period for all variables. Please be aware that because of an exaggeration of the solar radiation in the NORA3 atmosphere model (which was used for the simulation period 2012-20), we have identified a warm bias reaching up to 0.7-0.8 degrees in the surface layer in summer by performing a sensitivity test for 2021 where we had access to solar radiation estimates from both NORA3 and AROME MetCoOp. See Sec. 3.1. for references to the atmospheric models, and the same offset in sea temperature is also described in Gonzalez et al. (2025).

An additional question. You mention in the abstract that indeed the system you evaluate is the hindcast, but you want to use the system in forecast mode. There is no mention as to whether the forecasting system is expected to behave in the same way or not. Would it be worth clarifying?

Answer: Yes, we agree and have added text after the first paragraph in Sec. 6:

We aim at having a hindcast (reanalysis in the future) version that reflects the operational forecast version, however, an operational ocean model also depends on operational forcing data. A hindcast, however, dependent on the delay, has potentially easier access to better forcing data. The main purpose of the hindcast version is to allow users to access a long, consistent data series, but we also have better opportunities to test new forcing data, different numerics etc.

In figures showing salinity values there's no unit. Is this on purpose?

Answer: Thank you for pointing this out. We have added "psu" in the labels in Figure 7, 8 and 10.

I appreciated the colour diagrams shown in Fig 3, 4, 5, 6.

Answer: Thank you!

Just a suggestion not compulsory. For the metrics results in table 1 and 2, as you are lucky to have all these many metrics for quite a few locations, it would be really good to see them graphically. This could be done in a Taylor diagram.

Answer: Thank you for this suggestion. We agree and have displayed all coastal stations and the two depths (10 and 150m) for both temperature and salinity in one Taylor diagram. Table 1 and 2 is then replaced by a new Fig. 7 (Fig. 7, 8, 9 and 10 is now 8, 9, 10 and 11) with Figure caption:

Figure 7. Taylor diagram showing the correlation coefficient, normalized standard deviation, root-mean-square difference (grey curved lines) and normalized mean bias (bias divided by the observed standard deviation, indicated by symbol type and size) between modelled and observed temperature and salinity at 10 and 150 m depth at the seven fixed coastal hydrographic stations along the Norwegian coast for the period 2012-2023. Colors denote the different coastal stations. The two triangles below the horizontal axis represent salinity at 150 m depth at Sognesjøen and temperature at 150 m depth at Skrova shown with normalized standard deviation as the numerator and correlation coefficient as the denominator. The Taylor diagram is based on code from https://doi.org/10.5281/zenodo.15991044.

Sec. 4.2. (line 195-217) is rephrased to:

The hindcast archive has also been compared with the temperature and salinity profiles from the IMR fixed coastal stations for the period 2012-2023, and we have focused on the two vertical levels at 10 and 150 m depth, representing the surface layer and intermediate layer, respectively. Validation statistics are displayed for each station, depth level and parameter in a Taylor diagram (Fig. 7). The hydrographic properties in the surface layer are well reproduced by the model, with standard deviations close to the observed values and high correlations at all stations, although the agreement is better for temperature than for salinity. The normalized mean temperature biases are very low (<5%) at the southernmost stations, and somewhat higher at the northernmost stations and for salinity as the variability is lower. A complete time series from the station Ytre Utsira is displayed in Fig. 8, which demonstrates that the model closely follows the observations at this location which represents the southern part of the model domain. On shorter time and spatial scales, however, we can expect anomalously exaggerated water temperatures near the surface in the hindcast, and especially visible inshore.

The intermediate depth layer along the Norwegian coast is also reproduced realistically, though with slightly larger deviations between model and measurements. The validation statistics displayed in the Taylor diagram (Fig. 7) shows that the 150 m temperatures have realistic variability and generally high correlation with observations. An exception is the Skrova station, where the model overestimates the observed standard deviation and shows low correlation with observations. Similarly, the 150 m salinities at Sognesjøen are less well represented in Norkyst with relatively high standard deviations in the modeland low correlation coefficient. The latter low correlations are also seen in the intermediate layer salinities at Ytre Utsira, Eggum and Ingøy. A complete time serie from the station Skrova is shown in Fig. 9, and we see that while the model salinity is underestimated, the model temperature shows too high variability in the model.

**Technical corrections**

I would advise to correct a couple of typos throughout the document. The terms 'hydrographical' to be replaced by hydrographic, and 'horisontal' by horizontal.

Answer: We agree and have corrected these words accordingly.

---

## Author Comment (AC4)

**RC2**: 'Comment on egusphere-2025-3986', Stefania Angela Ciliberti, 10 Nov 2025

General comments: this paper presents a new implementation of the Norkyst v3 operational system in the very complex Norway coastal area. The system is based on ROMS whose model configuration and upstream data are described in Sections 2 and 3.

The paper could benefit from a dedicated section that introduces the validation methodology before going to Section 4, which is about results.
Answer: The introduction to Section 4 has been refined to make it more clear. Please see our reply to RC1.

Some references seem to be missing - for instance, upstream data used for the validation are mentioned but not referred, and that could help in the reproducibility.
Answer: We have included the missing references

Figures 7 and 8 might benefit from some additional work to improve readibility.
Answer: The figures have been refined and simplified.

Section 5 might need some improvement to better address the operational implementation of the system and the added value for users, including specific services at users' disposal for the access to forecast products and the associated product catalogue: the 2 subsections are quite short and I am sure the system is much more complex, so I would encourage the Authors to better balance this part of the paper, given its importance.
Answer: Since most of the details of the forecast system is similar to the hindcast system, they are not repeated. However, we have included some additional information and some examples of use of the operational products.

In the Conclusions, some key messages from the assessment could be reported and emphasized.
Answer: Yes, indeed. We have included some highlights from the validation.

An additional suggestion: intercomparison against available operational products in the area might significantly help to demonstrate the added value of Norkyst v3.
Answer: For our region it is only ARC MFC (i.e. "TOPAZ", which we are nesting into) that provides continuous forecast fields in addition to Norkyst, and TOPAZ lacks the necessary resolution for meaningful comparison especially in the fjords. Having said that, we are working on extending our validation and monitoring to include a side-by-side comparison of the shelf and open ocean dynamics, aiming to have it available through EDITO later this year. It is in our own interest too that TOPAZ should be as good as possible, of course, and we work closely with the TOPAZ developers.

Additional comments/suggestions are given in the supplement.
Answer: These comments are taken into account in the revised manuscript.